# Estimate the burden of sexual dysfunction due to non-communicable diseases in Ethiopia: Systematic review and meta-analysis

**Akine Eshete Abosetugn**[1]☺*, **Sisay Shewasinad Yehualashet**[2]☺

1 Department of Public Health, Debre Berhan University, Debre Brehan, Ethiopia, 2 Department of Nursing, Debre Berhan University, Debre Brehan, Ethiopia

☺ These authors contributed equally to this work.
* akine.eshete@yahoo.com

## Abstract

### Background

Untreated sexual dysfunction is a serious sexual problem that adversely affects the quality of life. Body of evidence indicates non-communicable diseases are common comorbid conditions associated with sexual dysfunction. Therefore, this review was aimed to synthesize and estimate the burden of sexual dysfunction and its determinant factors among patients with non-communicable diseases in Ethiopia.

### Methods

Cross-sectional studies were systematically searched using PubMed, Google Scholar, African Journals Online, Cochran Library, Scopus database, and gray literature. Data were extracted using a standardized Joanna Briggs Institute form. The $I^2$ statistic was used to check heterogeneity across the included studies. A funnel plot and Egger's regression test were used to check the presence of publication bias. Sensitivity analysis was deployed to check the effect of a single study on the overall estimation. All statistical analyses were done using STATA version 11.0 software.

### Result

A total of six studies with 2,434 study participants was included. The estimated pooled sexual dysfunction was 68.04% (95% CI: 56.41–79.67). Based on the subgroup analysis, the highest prevalence of sexual dysfunction was reported among patients with mental related illness, 73.02% (95% CI: 54.00–92.03).

### Conclusion

In this review, nearly seven out of ten patients with chronic illness have sexual dysfunction, which implies sexual dysfunction was highly prevalent among non-communicable patients. Therefore, health care providers should screen and manage sexual dysfunction during follow-up for improving patient quality of life and sexual reproductive health satisfaction.

**Editor:** Sina Azadnajafabad, Non-Communicable diseases Research Center, Endocrinology and Metabolism Population Sciences Institute, Tehran University of Medical Sciences, Tehran,Iran, ISLAMIC REPUBLIC OF IRAN

**Data Availability Statement:** All relevant data are within the manuscript and its Supporting information files.

**Funding:** The authors received no specific funding for this work.

**Competing interests:** The authors have declared that no competing interests exist.

**Abbreviations:** ED, Erectile dysfunction; JBI, Joanna Briggs Institute; NCDs, Non-Communicable Diseases; PRISMA, Preferred Reporting Items for Systematic Review and Meta-Analysis; SD, Sexual Dysfunction; SNNPR, Southern Nations Nationalities and People Region.

# Introduction

The burden of non-communicable diseases (NCDs) is one of the major health threats in the world. According to the World Health Organization (WHO's) report in 2018, NCDs killed 41 million people each year, accounting for 71% of all deaths in worldwide.

In developing countries, it is unacceptably high that contributed to 78% of deaths and 85% were premature deaths [1]. NCDs have been steadily increasing and contributing to 39.3% of deaths. Now a day NCDs have been becoming a major agenda in Ethiopia [2, 3].

Sexual dysfunction is a comorbidity of NCDs in both sexes [3, 4]. Evidence showed that sexual dysfunction was an organic complication associated with diabetes mellitus [5], cardiovascular diseases [6], hypertension [7], and stroke [8]. Unmet needs for sexual and reproductive health services in both sexes are compounded by the increased burden of noncommunicable diseases that negatively impact women's [9] and men's reproductive health [10]. Untreated sexual dysfunction can have major consequences for patients, as it decreases their quality of life and self-esteem, and is linked to a decrease in sexual satisfaction [11, 12].

Sexual dysfunction (SD) is a sexual behavior and sensation disorder that manifests as a lack of sexual psychology and physiological response [13]. It is any physical or psychological problem that prevents the person or couple from getting sexual satisfaction from sexual activity [14]. SD is a general term that includes erectile dysfunction, failure of sexual intercourse, and loss of desire [13].

In Ethiopia, SD is a common yet underappreciated complication of most NCDs. The global burden of SDs in both sexes has been reported in several studies, ranging from 43.8% to 85.5% [15–21]. Despite the fact that these studies differed, they all demonstrated that SD is considered a public health issue. However, these large variances throughout Ethiopia's geographical areas may make it difficult for planners, policymakers, implementors, and service providers to create trustworthy evidence. As a result, this systematic review and meta-analysis was to synthesize and estimate the burden of SD and its determining factors among patients with noncommunicable diseases in Ethiopa. The goal of this systematic review is to find out "what is the estimated burden of SD among NCD patients in Ethiopia?". Finally, the findings of this systematic review and meta-analysis will be utilized to update planners and policymakers on how to improve patient and health-care provider communication in order to overcome SD. It was also used as a starting point for researchers looking for potential causes of SD in NCD patients.

# Methods

## Study settings

This systematic and meta-analysis study included studies done only in Ethiopia. Ethiopia is one of the east African countries in the Horn of Africa. It covers 1.104 million $km^2$ and is divided into nine regions specifically Tigray, Afar, Amhara, Oromia, Somali, Benishangul-Gumuz, Southern Nations, Nationalities, and People Region (SNNPR), Gambella, Harari, and two Administrative states (Addis Ababa town administration and Dire Dawa town administration).

## Inclusion criteria

**Type of studies.**   Cross-sectional studies and government reports on sexual dysfunction caused by non-communicable diseases were considered, both published and unpublished. Human subjects, full-text articles, gray and grey literature authored in English and published in peer-reviewed journals between 2000 and 2020 were included.

**Study participants.** Patients with any types of NCDs, including diabetes, hypertension, cancer, cardiovascular disease, and mental illness.

**Types of outcome measures (sexual dysfunction).** A standardized tool was used to assess sexual dysfunction. The International Index of Erectile Function (IIEF-5) consists of five items, each of which was assessed on a five-point ordinal scale, with lower scores indicating poorer sexual function. As a result, a question response of 1 was deemed the least functional, while an answer of 5 was deemed the most functional. The IIEF-5 offers a range of possible scores from 1 to 25 (one question has a range of 1–5), with a score of 21 or higher indicating normal erectile function and a score of 21 or lower indicating Erectile Dysfunction (ED). On the basis of IIEF-5 scores, ED is divided into four categories: severe (1–7), moderate (8–11), mild to moderate (12–16), mild (17–21), and no ED (22–25) [22].

The total score for both the female and male based on the Changes in Sexual Functioning Questionnaire (CSFQ-14) was calculated by adding the values of items 1 to 14. Scores of 41 or less for females and 47 or less for men showed sexual dysfunction [23].

**Exclusion criteria.** Citations without abstracts and/or full text, commentaries, anonymous reports, letters, duplicate studies were excluded.

## Search strategy

Initially, the Cochrane database of a systematic review, Joanna Briggs Institute (JBI) database of a systematic review and PROSPERO databases were checked for the presence of ongoing studies related to the current topic.

Electronic database searches were conducted using PubMed/PMC, Google Scholar, African Journals Online, Scopus database, and Cochrane Library from May 1-31/ 2020, and updated June 15/2021 to decrease time-lag bias. To check the availability of articles on institutional repositories, a manual search was undertaken from gray literature. To find relevant articles, we looked through the reference lists of all the articles we found. The search was organized using CoCoPop, with the following parameters: context (Ethiopia), condition (Sexual dysfunction related to NCDs), and population (patient with non-communicable diseases).

Using Boolean operators, the search strategies were developed. The following search strategy was applied.

| | MeSH Terms | Article yield |
|---|---|---|
| First search | ("sexual dysfunctions, psychological"[MeSH Terms] OR ("sexual"[All Fields] AND "dysfunctions"[All Fields] AND "psychological"[All Fields]) OR "psychological sexual dysfunctions"[All Fields] OR ("sexual"[All Fields] AND "dysfunction"[All Fields]) OR "sexual dysfunction"[All Fields] OR "sexual dysfunction, physiological"[MeSH Terms] OR ("sexual"[All Fields] AND "dysfunction"[All Fields] AND "physiological"[All Fields]) OR "physiological sexual dysfunction"[All Fields] OR ("sexual"[All Fields] AND "dysfunction"[All Fields])) AND ("ethiopia"[MeSH Terms] OR "ethiopia"[All Fields]) | 639 |
| 2nd search | Burden[All Fields] AND ("noncommunicable diseases"[MeSH Terms] OR ("noncommunicable"[All Fields] AND "diseases"[All Fields]) OR "noncommunicable diseases"[All Fields] OR ("non"[All Fields] AND "communicable"[All Fields] AND "disease"[All Fields]) OR "non communicable disease"[All Fields]) AND ("sexual dysfunctions, psychological"[MeSH Terms] OR ("sexual"[All Fields] AND "dysfunctions"[All Fields] AND "psychological"[All Fields]) OR "psychological sexual dysfunctions"[All Fields] OR ("sexual"[All Fields] AND "dysfunction"[All Fields]) OR "sexual dysfunction"[All Fields] OR "sexual dysfunction, physiological"[MeSH Terms] OR ("sexual"[All Fields] AND "dysfunction"[All Fields] AND "physiological"[All Fields]) OR "physiological sexual dysfunction"[All Fields] OR ("sexual"[All Fields] AND "dysfunction"[All Fields])) AND ("ethiopia"[MeSH Terms] OR "ethiopia"[All Fields]) | 82 |

## Data extraction

The essential data were extracted using a standardized data extraction form developed by the Joanna Briggs Institute (JBI). The authors' names, year of publication, study area, study design, sample size, a measurement for sexual dysfunction, prevalence of sexual dysfunction, and information about the domain of sexual dysfunction are all included in the data extraction format.

Two authors (AE, SS) were independently extracted all relevant data from each study. During the extraction process, data discrepancy among data extractors was resolved by discussion and consensus and also by referring back to the original study. The PRISMA flow chart was used to summarize the screening and selection procedure for the reviewed articles [24].

## Risk of bias

To remove duplicate studies, the data was transferred to Endnote version 7. A search strategy was developed by the investigators (AE and SS). The reviewers were blinded to the journal, authors, and results. There were no conflicts between the two reviewers in the final decisions. The selections of identified studies were done in two stages. In the first stage, a selection of relevant studies is based on titles and abstracts. In the second stage, studies that met the inclusion criteria and the full paper were found for detailed assessment.

Using JBI criteria [25], two authors independently assessed articles eligibility and risk of bias for the included studies. JBI developed a critical appraisal checklist for cross-sectional studies to evaluate methodological quality and risk of bias ("S1 Table").

## Data synthesis and statistical analysis

The STATA software version 11.0 was used to analyze the data. A random-effects model is used to display the extracted data from each study in a meta-analysis. A forest plot was used to generate the pooled effect size of sexual dysfunction with a 95% confidence interval (CI).

Heterogeneity between studies were assessed using the Cochran's Q and $I^2$ statistic [26] and $I^2$ statistical test with a value of $\geq 75\%$ and $p \leq 0.05$ indicating the presence of moderate to high levels of significant heterogeneity [27]. Subgroup analysis and meta-regression analysis were performed to investigate sources of heterogeneity. In a sensitivity analysis, potential individual outliers were treated by deleting each study one at a time. To determine the presence of publication bias, the funnel plot and Egger's test were utilized [28]. The Egger test revealed a statistically significant publication bias with a p-value less than 0.05. To account for the impacts of publication bias, trim-and-fill analysis was performed.

## Data presentation and reporting of results

PRISMA [29] form was used to report the findings. A PRISMA flow diagram was used to summarize the screening and selection procedure for the examined articles.

# Results

## Search results

The search strategy retrieved 823 studies from PubMed, Cochrane library, Google scholar, and gray literature. Of these, 13 articles were searched manually using grey literature. After removing duplicate publications, 205 articles remained. About 198 articles were excluded by reading the titles and abstracts of the studies based on the pre-defined eligibility criteria. Seven articles were included for systematic review and screened for further assessment. Finally, six studies were included for meta-analysis (Fig 1).

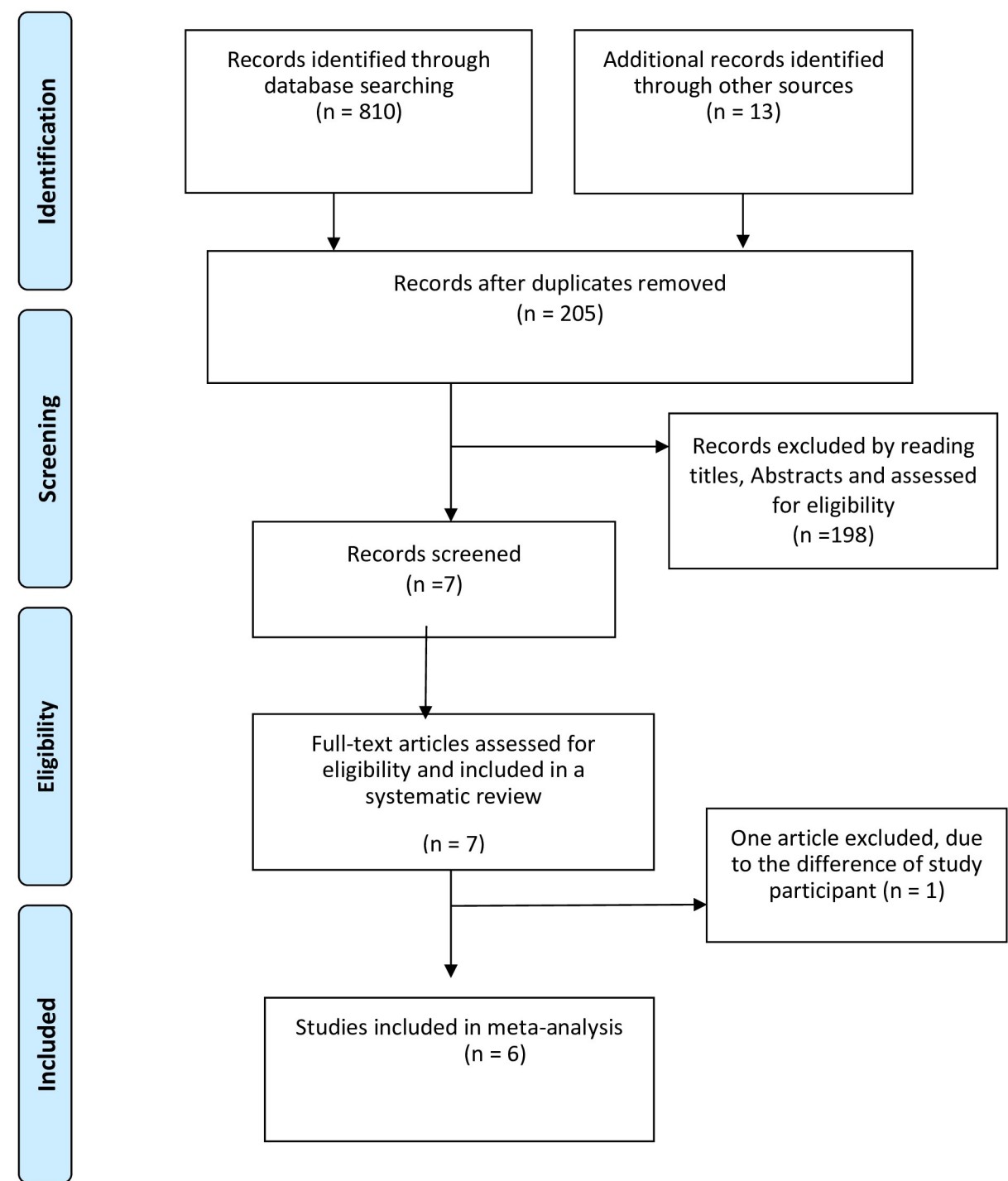

**Fig 1. The PRISMA flow diagram of identification and selection of studies for the systematic review and meta-analysis.**

## Characteristics of included studies

For a systematic review, the search approach provides a total of seven relevant papers with a total sample size of 3279. The samples of each article were varied significantly in size, ranges from 249 to 845. Two studies were conducted in the Amhara region [15, 16], two in the Tigray

**Table 1. Description of the included studies.**

| Authors, with a publication year | Study area/ region | Sample | Prevalence of sexual dysfunction (SD), n (%), and other important findings | Type of NCDs | Type of measurement | JBI-Quality score |
|---|---|---|---|---|---|---|
| Ejigu AK et al., 2019 [32] | Addis Ababa | 576 | • 363(63.3%) had sexual dysfunction<br>• SD for male 67.4%, for female 55.6%<br>• Frequency of SD domains; sexual arousal problem, 97.8%, sexual pain problem, 11.3%, sexual desire problem, 94.8%, orgasmic dysfunction 89.9%, sexual pleasure problem 95.1% | Epilepsy | CSFQ-14- | 8/ 9*100 = 89% |
| Asefa A. et al., 2019 [21] | SNNPR | 398 | • 212 (53.3%) had SD<br>• SD for male 52%, for female 56.6%<br>• Frequency of SD domains; sexual arousal problem, 40.2%, sexual desire problem, 48.2%, orgasmic dysfunction, 45.7%, sexual pleasure problem, 34.4%, desire frequency problem 55.8% | Diabetes mellitus | CSFQ-14 | 8/ 9*100 = 89% |
| Fanta T. et al., 2018 [19] | Addis Ababa | 422 | • 349 (82.7%) had SD<br>• SD for male 84.5, for female 78.6 | Schizophrenia | CSFQ-14 | 8/ 9*100 = 89% |
| Walle B. et al. 2018 [16] | Amhara | 422 | • 361 (85.5%) had SD<br>• Category of ED: Mild ED, 66(18.3), Moderate ED, 234(34.18%) and severe ED, 61(16.9) | Diabetes mellitus | IIEF-5 | 8/ 9*100 = 89% |
| Seid A. et al. 2017 [30] | Tigray | 249 | • 174 (69.9%)) had SD<br>• Category of ED: Mild ED, 66(18.3%), Moderate ED, 234(34.18%) and severe ED, 61(16.9%) | Diabetes mellitus | IIEF-5 | 8/ 9*100 = 89% |
| Tesfaye et al. 2020 [15] | Amhara | 367 | • 195 (53.1) had SD | Diabetes mellitus | IIEF-5 | 8/ 9*100 = 89% |

Changes in Sexual Functioning Questionnaires (CSFQ-14-), International index of erectile function (IIEF-5), SNNPR; Southern Nations Nationalities and People Region, ED; Erectile dysfunction, SD; Sexual Dysfunction.

region [30, 31], two in Addis Ababa [19, 32], and one study in the Southern Nation Nationality People Region (SNNPR) [21]. Six studies were institution-based cross-sectional studies, whereas one study [31] was a community-based cross-sectional study. The quality of each study was scored based on the JBI quality assessment checklists. The main features including studies were presented in table (Table 1).

## The burden of sexual dysfunction

The highest prevalence of sexual dysfunction was observed in Amhara (85.5%) [16] and Addis Ababa (82.7%) [19], whereas the lowest prevalence was in Tigray (43.8%) [31]. The highest burden of sexual dysfunction was associated with diabetes mellitus and schizophrenia patients [16, 19].

The sexual dysfunction among males was reported in both studies conducted in Addis Ababa (82.7%) [19], and (67.4%) [32], whereas in females, sexual dysfunction was observed in Addis Ababa (78.6%) [19]. In all included studies: sexual arousal dysfunction, sexual pain problem, pleasure dysfunction, sexual desire disorder, orgasmic dysfunction, desire dysfunction were the most prevalent sexual dysfunctions.

Erectile dysfunction was a problem for the patients. In a study conducted in Amhara, the prevalence of erectile dysfunction was high (85.5%) [16]. The majority of the patients [16, 30] had moderate erectile dysfunction. In one study, severe erectile dysfunction was found 16.9% [16].

## Meta-analysis results

**The estimate of sexual dysfunction.** The estimated pooled sexual dysfunction among NCDs patients using the random effect model was 68.04 (95%CI: 56.41–79.67) with

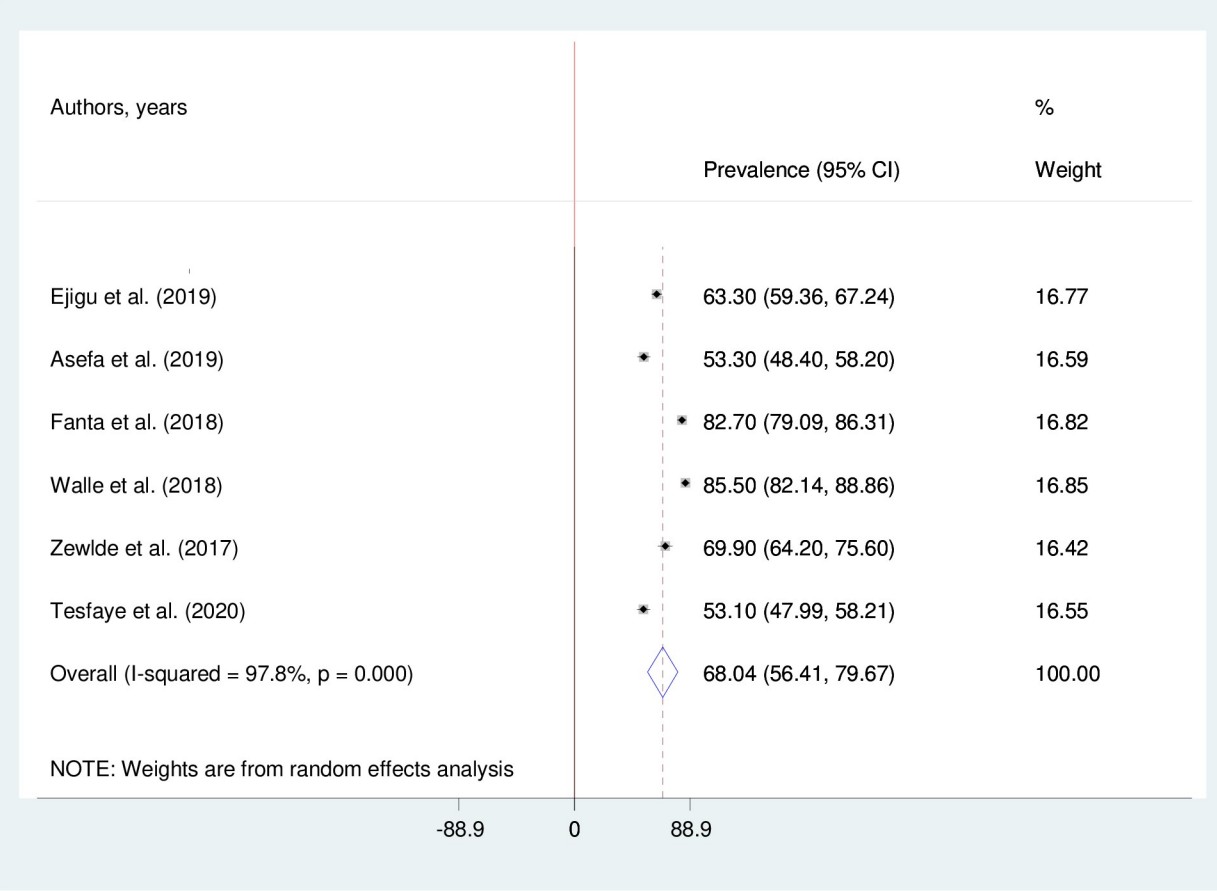

**Fig 2. The pooled prevalence of sexual dysfunction in patients with non-communicable disease in Ethiopia.**

heterogeneity ($I^2$ = 97.8%; p < 0.001). The weights of studies using the random-effect model were ranging from 16.42 to 16.82 (Fig 2).

**Subgroup analysis.** The study area, publication year, and type of NCDs were used to create subgroups. There was variability among the included studies in all subgroup analyses (Table 2).

**Meta-regression analysis.** A meta-regression analysis of the study area, year of publication, type of NCDs, and sample size found that the number of covariates adjusted in the analysis was not associated with the estimated OR (P = 0.887, P = 0.149, P = 0.634, P = 0.972) respectively (Table 3).

**Sensitivity analysis.** Sensitivity analysis was performed, one study is excluded at a time and the impact of removing each of the studies is evaluated on the pooled estimate and heterogeneity. Sensitivity analyses using the random-effects model revealed that no single study influenced the estimates (Fig 3).

**Publication bias.** Regarding publication bias, the visual inspection of the funnel plot was asymmetrical at the bottom (Fig 4), but the Egger test showed that a p-value of 0.112 which indicated no evidence of publication bias. The trim-and-fill method imputed for missing studies and recalculated our pooled but no significant change in the finding (Fig 5).

**Table 2. Subgroup analysis by different category of the studies.**

| Categories | Sexual dysfunction with (95% CI) | Heterogeneity | No. studies |
|---|---|---|---|
| **Study area / Regional status** | | | |
| • Addis Ababa | 73.02 (54.00–92.03) | I2 = 98.0%, p = 0.000 | 2 |
| • Amhara region | 69.36 (37.61–101.11) | I2 = 99.1%, p = 0.000 | 2 |
| • SNNPR and Tigray | 61.53 (45.27–77.80) | I2 = 94.7%, p = 0.000 | 2 |
| **Year of publication** | | | |
| • >2019 | 56.71 (49.64–63.78) | I2 = 85.7%, p = 0.000 | 3 |
| • <2017 | 79.71 (71.96, 87.47) | I2 = 90.8%, p = 0.000 | 3 |
| **Type of NCDs** | | | |
| • Mental health-related illness (Epilepsy and schizophrenia) | 73.02 (54.00, 92.03) | I2 = 98.0%, p = 0.00 | 2 |
| • Diabetes mellitus | 65.50 (48.09, 82.91) | I2 = 98.2%, p = 0.000 | 4 |

**Table 3. Meta-regression analysis for the included studies to identify the source(s) of heterogeneity.**

| Variable | Coef. | Std. Err. | P>|t| | 95% CI |
|---|---|---|---|---|
| Study area/region | -0.02577 | 0.1701 | 0.887 | (-0.497,0.4463) |
| Publication year | -0.1946 | 0.109 | 0.149 | (-0.497, 0.108) |
| Type of NCDs | -0.14678 | 0.285 | 0.634 | (-0.938, 0.645) |
| Sample size of study | 0.001 | 0.00014 | 0.972 | (-0.0039, 0.0041) |

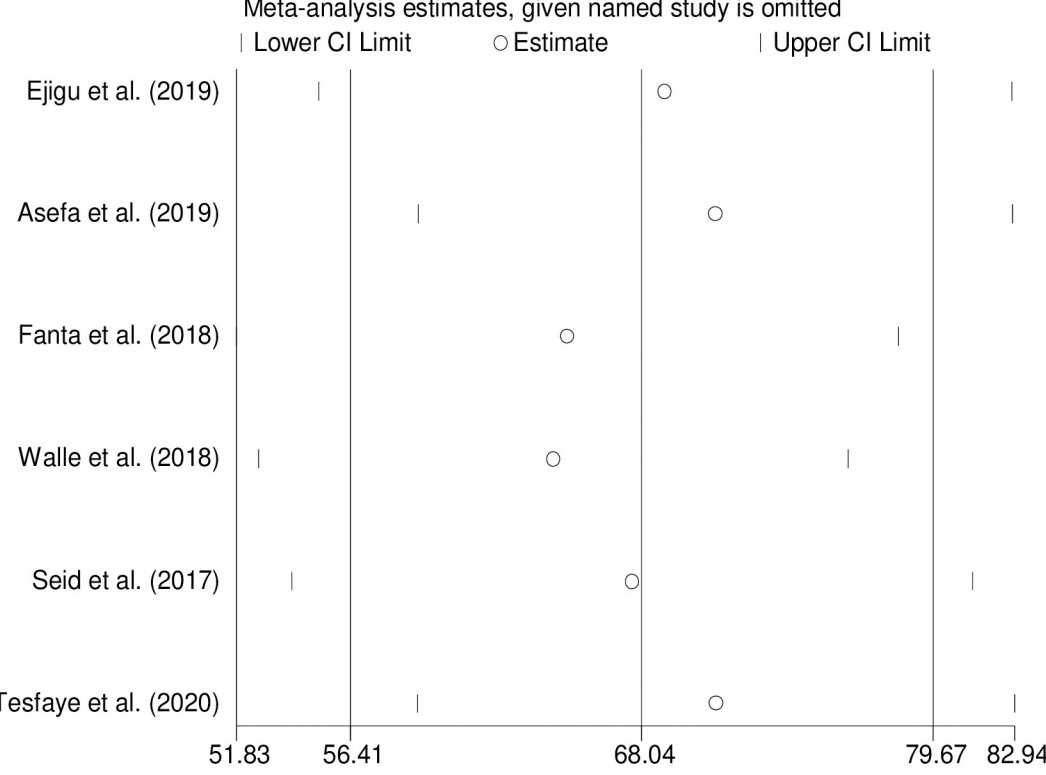

**Fig 3. Plot of sensitivity analysis to assessing the influence of individual study.**

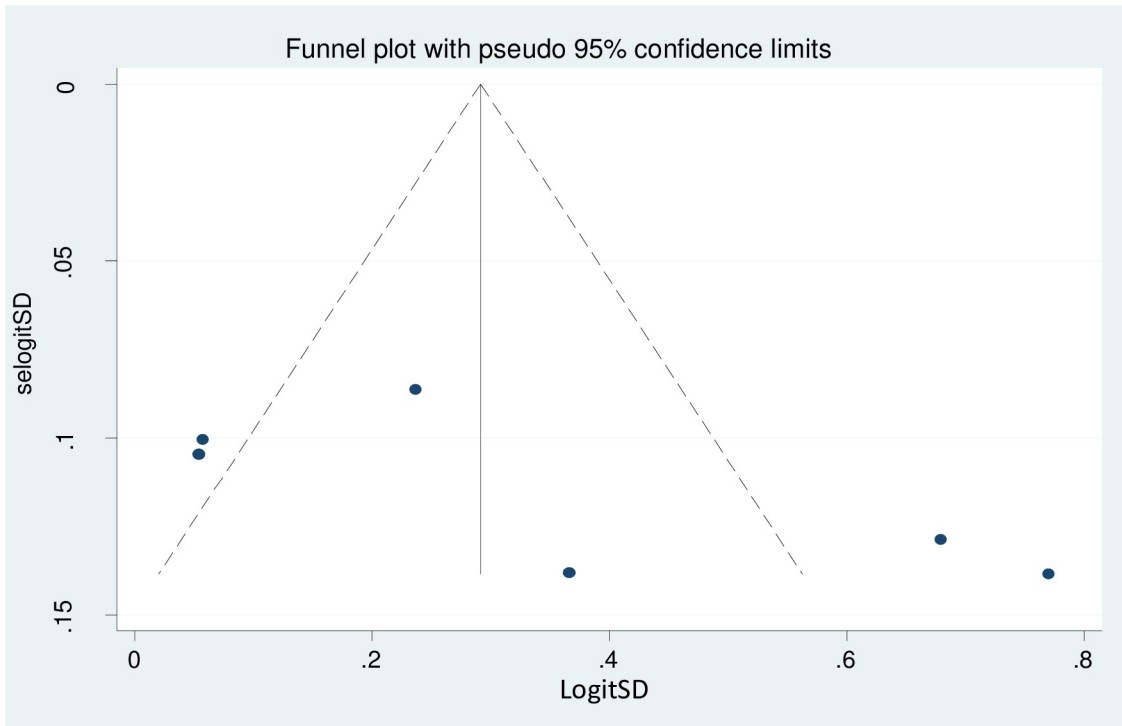

**Fig 4. Funnel plot of effect estimates against standard error of log estimate.**

## Discussion

Sexuality is one of the facets of quality of life and wellbeing [33, 34]. However, sexual functioning and sexual satisfaction rarely consider aspects of the life of patients with NCD in clinical practice because NCD assessment and treatment typically focus on the reduction of symptoms. Little attention is dedicated to the strengths and positive outcomes of individuals with this condition.

This systematic review and meta-analysis was to synthesise and quantify the pooled level of sexual dysfunction among NCDs patients in Ethiopia. Sexual dysfunction was shown to be prevalent among non-communicable disease patients, ranging from 43.8% [31] to 85.5% [16]. Sexual arousal dysfunction, sexual pain problem, pleasure dysfunction, sexual desire disorder, orgasmic dysfunction, and desire dysfunction were the most common sexual dysfunctions in both sexes, according to the study. Understanding the barriers to sexual dysfunction associated with noncommunicable disease complications was therefore crucial in developing effective policies, programs, and interventions that were appropriate for the nature of the sexual disorder.

In this current meta-analysis, the estimated pooled sexual dysfunction was 68.04% with 95% CI: (56.41–79.67) among patients with non-communicable patients in Ethiopia. This pooled estimate of sexual dysfunction was nearly consistent with a systematic and meta-analysis study in the world 76% [35] and a systematic and meta-analysis study in Africa 71.45% [36]. As a result, this meta-analysis finding suggests that sexual dysfunction is a common but often neglected consequence among non-communicable disease patients. To manage sexual dysfunction as a result of NCDs, a multimodal therapeutic approach must be strengthened, with a focus on treatment adherence and psychological support.

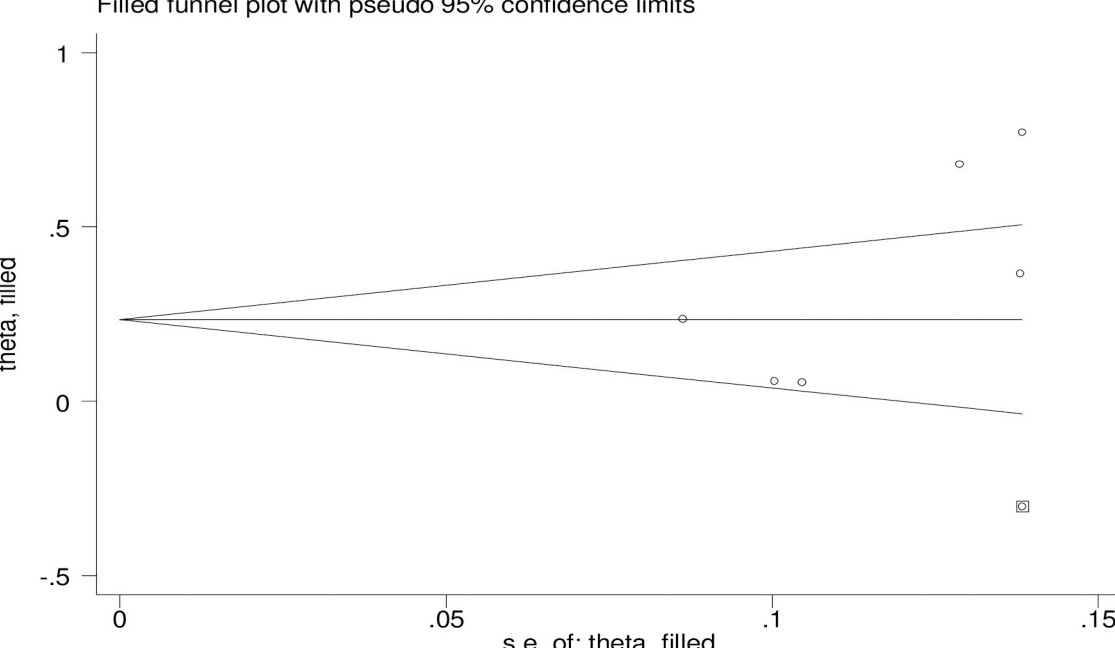

**Fig 5. Result of trim and filled analysis for adjusting publication bias.**

Despite the fact that non-communicable disease patients have a high rate of sexual dysfunction, the majority of them do not seek medical help [37]. Researchers cited embarrassment, shame, stigmatization fear, and a threat to marriage as reasons for not reporting to doctors [37, 38]. On the other side, physicians are reluctant to engage with their patients on issues of sexual functioning due to socio-cultural barriers [39]. As a result, the unmet needs for sexual and reproductive health needs are coupled with the rising burden of non-communicable diseases that adversely affect the reproductive health of women [9] and men [10]. Indeed, untreated sexual dysfunction is strongly linked to a patient's quality of life and self-esteem, both of which are directly linked to sexual satisfaction [11, 12].

Based on subgroup analysis, the pooled prevalence of sexual dysfunction among patient with mental related illness (particularly epilepsy and schizophrenia) was 73.02% (95% CI: 54.00–92.03). Several mental disorders and related treatments interfere with sexual function [40]. Several prospective cohort studies have explored the relation between sexual dysfunction and the risk of depression [41, 42]. The prevalence of sexual dysfunction may be explained by comorbid depression due to overlapping symptoms such as "loss of interest and pleasure feelings" and "loss of libido". As a result of this finding, clinicians can better design preventative strategies and therapies to increase patient quality of life and sexual satisfaction.

This systematic review and meta-analysis have different clinical and public health practice implications. Estimating the prevalence of sexual dysfunction among patients with noncommunicable diseases is critical for guiding healthcare professionals in minimizing NCD-related sexual satisfaction complications. Furthermore, it provides data on the burden and public health impact of sexual dysfunction caused by non-communicable diseases in a country, which can be used to improve health-care policy and clinical practice.

There are some limitations to this systematic review and meta-analysis study that should be considered in future research. First, because there are so few studies, it may be difficult to generalize the results to all NCD patients in the country. Second, in the included studies, different

criteria for diagnosing sexual dysfunction were used, which could alter the assessment of the pooled prevalence of sexual dysfunction. Third, this study does not identify the barriers to sexual dysfunction among NCD patients or the factors that contribute to it.

Ethiopia, being one of Africa's most populous countries, may suffer a burden of early deaths and disabilities due to NCDs by 2040. Even though, the national response to NCDs remains fragmented and insignificant, there are encouraging actions on NCDs in terms of political commitment, NCDs prevention, and treatment services at the primary health care level [43].

In order to improve a patient's quality of life, screening and management of sexual dysfunction must be included as part of the assessment throughout their follow-up [44, 45]. These findings may have clinical implications, such as practical suggestions for the prevention and management of sexual dysfunction in NCD patients. A physician and health care provider should diagnose sexual dysfunction in NCD patients as soon as possible. Furthermore, the national health system must be reorganized to assure acknowledgement of the NCD burden, maintain political commitment, allot sufficient financing, and improve the organization and delivery of NCD services related to reproductive and sexual health at the primary health care level.

Despite the fact that pathophysiological variables are not yet identified in this investigation. This demonstrated that all-cause sexual dysfunction could reflect the existence of common pathogenic pathways among NCDs patients, therefore more research is needed to identify other risk variables such as disease duration or other pathological variables such as aging, inflammation, medication, and stress in the development of sexual dysfunction across a wide variety of illnesses. Furthermore, comprehending the barriers to sexual dysfunction associated with non-communicable disease complications was critical in developing context-based specific preventive strategies, as well as treatment focusing on adherence and psychological support for improving patient quality of life and sexual reproductive health satisfaction.

Unhealthy diets, physical inactivity, cigarette use, and alcohol abuse are the key risk factors for NCDs. As a result, the majority of these diseases are preventable, as they proceed early in life as a result of lifestyle factors [46]. In the public health sector, there is growing concern that inadequate eating has raised the risk of chronic diseases and nutrition difficulties [47]. As a result, the primary prevention of diseases focuses lifestyle modifications and interventions to reduce the risk of NCDs.

The NCD prevention plan is based on risk factor management, which includes resource allocation, multi-sectoral collaborations, knowledge and information management [48, 49]. Monitoring and assessing the progress of NCDs at the national, regional, and global levels is important.

NCDs are the primary health-care challenge in modern society. The management of risk factors is important in the treatment of NCDs. NCD required to deliver a variety of interventions from many perspectives and at various levels, including individual and national levels. Based on the assumptions developed throughout the scientific discussion above, it can be stated that modern strategies for the management of NCDs should be focused toward the individual level, where the individual is responsible because of their own health by simply living a healthy lifestyle.

To reduce NCD risk factors for disease burden and adverse outcomes, multi-sectoral activities are required. The findings will help policy and programming to reduce the burden of NCDs in the long run. Cultural norms may have an impact on how physicians manage patients with sexual problems in Ethiopia, where discussing sexual problems is taboo.

If we want to improve the treatment of sexual dysfunction caused by NCDs, we must first improve screening. Screening is not the only poor relation to the sexual health of patients suffering from NCDs. There are no specific scales allowing concrete evaluation of the perception

of good health or good sexual health for these patients. Currently there are no well-structured protocols for treatment of sexual difficulties.

## Conclusion

According to this systematic review and meta-analysis, sexual dysfunction was found to be 68.04% in Ethiopia, with a 95% confidence interval of (56.41–79.67). This means that roughly seven out of 10 non-communicable illness patients in Ethiopia experience sexual dysfunction, implying that sexual dysfunction is very common among non-communicable diseased patients. The combined prevalence of sexual dysfunction among individuals with mental illnesses (especially epilepsy and schizophrenia) was 73.02%t among all NCDs (95% CI: 54.00–92.03).

## Supporting information

**S1 Table. JBI critical appraisal checklist for studies reporting prevalence data.**
(DOCX)

## Acknowledgments

We acknowledge the Authors of each article for reviewing their article. We would like to thank JBI for using their systemic review and meta-analysis guidance.

## Author Contributions

**Conceptualization:** Akine Eshete Abosetugn.

**Data curation:** Akine Eshete Abosetugn, Sisay Shewasinad Yehualashet.

**Formal analysis:** Akine Eshete Abosetugn, Sisay Shewasinad Yehualashet.

**Investigation:** Akine Eshete Abosetugn.

**Methodology:** Akine Eshete Abosetugn.

**Project administration:** Akine Eshete Abosetugn.

**Supervision:** Akine Eshete Abosetugn.

**Validation:** Akine Eshete Abosetugn.

**Writing – original draft:** Akine Eshete Abosetugn, Sisay Shewasinad Yehualashet.

**Writing – review & editing:** Akine Eshete Abosetugn, Sisay Shewasinad Yehualashet.

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
