## [Decision Letter · Decision Letter 0]

29 Sep 2020

PONE-D-20-18974

The Burden of Non-Communicable Disease on Sexual Dysfunction in Ethiopia: systematic review and meta-analysis

PLOS ONE

Dear Dr. Eshete,

Thank you for submitting your manuscript to PLOS ONE. After careful consideration, we feel that it has merit but does not fully meet PLOS ONE’s publication criteria as it currently stands. Therefore, we invite you to submit a revised version of the manuscript that addresses the points raised during the review process.

We look forward to receiving your revised manuscript.

Kind regards,

Claudia Marotta

Academic Editor

PLOS ONE

Additional Editor Comments:

Dear authors, follow reviewers suggestions to improve your manuscript

Journal Requirements:

Reviewers' comments:

Reviewer's Responses to Questions

**Comments to the Author**

1. Is the manuscript technically sound, and do the data support the conclusions?

Reviewer #1: Partly

Reviewer #2: No

Reviewer #3: Yes

2. Has the statistical analysis been performed appropriately and rigorously? 

Reviewer #1: Yes

Reviewer #2: I Don't Know

Reviewer #3: I Don't Know

3. Have the authors made all data underlying the findings in their manuscript fully available?

Reviewer #1: Yes

Reviewer #2: Yes

Reviewer #3: Yes

4. Is the manuscript presented in an intelligible fashion and written in standard English?

Reviewer #1: No

Reviewer #2: No

Reviewer #3: No

5. Review Comments to the Author

Reviewer #1: Question 1.

The paper by Eshete and Shewasinad reviewed sexual dysfunction (SD) among patients diagnosed with non-communicable diseases (NCD) in Ethiopia, synthesized and came up with an estimated burden of SD among patients with NCDs.

However, this differs from the objective of the study as stated by the authors which was to synthesize, and estimate the burden of non-communicable disease on SD and its determinant factors.

Based on the findings, the objective of the study ought to have been: to synthesize and estimate the burden of SD and its determinant factors among patients with non-communicable diseases in Ethiopia.

Secondly, the study did not find the ‘determinant factors’ stated in the objective. I suggest this aspect to be expunged.

Thirdly, based on the findings, the research question ought to have been: ‘What is the estimated burden of sexual dysfunction among patients with NCDs in Ethiopia’?

Fourthly, as stated by the authors under limitations, it will be difficult to generalize on the whole country based on just 6 papers! Furthermore, are the authors looking at NCD as a risk factor for sexual dysfunction, if so, there will be need for further studies to determine other risk factors for SD and compare them with those obtained elsewhere? Further studies should also aim at understanding the perspectives of the NCD patients found to have SD…. which of the two began first, did they happen at the same time or are they merely coincidental?. Qualitative studies will best answer these pertinent questions going forward.

Question 2: Yes

Question 3: Yes

Question 4.

I recommend language editing of the whole document as there are several typo graphical and grammar errors

Under the sub heading 'implications of the study findings', the first factual sentence 'Ethiopia will be the first among the most populous nations in Africa to experience the dramatic burden of premature deaths and disability from NCDs by 2040' should be backed by a reference.

For tables, the heading should be in italics and on top

For figures, the heading should be normal, non-italicized and below

Reviewer #2: The authors of this study made a precious effort to investigate the burden of non-communicable diseases (NCDs) on sexual dysfunction in Ethiopia, especially focusing on the quality of life of patients with NCDs. This systematic review and meta-analysis tried to investigate the mentioned notion through a comprehensive review and statistical analyses. However, some comments are necessary to be addressed and I provided them in the uploaded attachment file.

Reviewer #3: The manuscript reports on the prevalence of sexual dysfunction among patients with noncommunicable diseases in Ethiopia. While the manuscript provides important information, there is need for improvement of the manuscript. The authors may also consider revising the title of the manuscript.

The language needs significant improvement. Words are used inappropriately and sometimes incorrectly throughout the manuscript. There are several sentences which are unclear, with some sentences being very long. The wrong tense is used in several areas of the manuscript. Sentence construction needs to be improved in many areas; some of the phrases used are incomprehensible. There is some repetition; there is at least one instance in which the same sentence is used in different areas of the manuscript.

Acronyms are used without explanation, in the text JBI and in Tables (SD, ED). Some terms are being used interchangeably in the text – sexual dysfunction and erectile dysfunction, even though they do not have the same meaning.

There are areas where the text does not match what is presented in Table I. Numbers are sometimes quoted incorrectly and sometimes not listed

…the sexual dysfunction among males was reported in both studies conducted in Addis Ababa (82.7% [19] The SD listed for males for the said study is 84.5 in Table 1 and the SD for male in the second study [20} is not listed in the text

.….whereas the lowest prevalence was in Tigray (43.8%)[18]. That study is not listed in Table 1.

…………whereas in females, sexual dysfunction was observed in Addis Ababa( 78.6%) [19] There is another study done in Addis Ababa and listed in Table 1: Ejigu AK et al 2019 [20] also reported sexual dysfunction in females 55.6%, but this is not included in the text.

Although 7 studies are being cited in the text, only 6 studies are presented in Table 1. The Abstract also states 6 studies.

There is inconsistency in the way 96% CI are written in the text and Tables.

Sexual dysfunction and erectile dysfunction are being used interchangeably in the text.

Incorrect conclusions are made: The conclusion incorrectly states …..that sexual dysfunction in Ethiopia was 68.04%....without indicating that this was in patients with NCDS

The limitations section needs review. The authors seem unaware that the study limitations also need to be taken into consideration when conclusions are made in relation to this study.

Although the authors indicate that there was a small number of studies in the systematic review and meta-analysis, strong statements are made with no references provided: This indicates that seven out of ten patients with non-communicable disease have sexual dysfunction, which implies sexual dysfunction was highly prevalent among non -communicable (disease missing) patients in Ethiopia.

The Figures in the manuscript are not numbered sequentially.; 1,2, 3,5,6. There is an error in Figure 1: Records screened (n=7).

Implications of the Study Findings: This segment needs significant revision. A reference needed for the statement made in the 1st sentence. Very long unclear sentences are used in the final paragraph. Overall this section lacks clarity.

Please note that I am unable to comment on the statistical analyses used in this study. This is because I am not familiar with statistical analyses which are used for systematic reviews and meta-analyses.

6. PLOS authors have the option to publish the peer review history of their article (what does this mean?). If published, this will include your full peer review and any attached files.

Reviewer #1: **Yes: **Haruna Ismaila Adamu, MBBS; MPH; PhD

Reviewer #2: **Yes: **Sina Azadnajafabad

Reviewer #3: No

---

## [Author Response · Author response to Decision Letter 0]

17 Feb 2021

Dear Academic Editor!

PLOS ONE 

Response to Reviewers

I am happy to resubmit for publication version of “Estimate the burden of sexual dysfunction due to non-communicable disease in Ethiopia: systematic review and meta-analysis” for a review as original research in PLOS ONE.

The comments of the editor and the reviewers were highly insightful and enabled us to greatly improve the quality of our manuscript. Therefore, based on the editor’s and the reviewers’ concerns we have made extensive edition in our manuscript. Particularly, we have extensively edited the manuscript by a professional language editor (English-language instructors of Debre Berhan University thoroughly edited the manuscript for language usage, spelling, and grammar) before submitting the revised version. The formatting of the text and document (text sizes and grammatical errors) was also edited. We have addressed yours’ concerns in a point by point format. 

 We look forward to hearing from you at your earliest convenience. 

Thank you for your consideration of this manuscript! 

Kind regards,

Akine Eshete

On behalf of authors

The authors of this study made a precious effort to investigate the burden of non-communicable diseases (NCDs) on sexual dysfunction in Ethiopia, especially focusing on the quality of life of patients with NCDs. This systematic review and meta-analysis tried to investigate the mentioned notion through a comprehensive review and statistical analyses. However, some comments are necessary to be addressed and I provide them below:

1. The title of the manuscript is not fully appropriate and consistent with the study aim. The last sentence of the background part of the abstract “estimate the burden of sexual dysfunction due to non-communicable disease in Ethiopia” is more appropriate for such investigation.

Response: We thank you very much for this important recommendation. We edited the title accordingly.

2. “Non-communicable disease” is better to be replaced by “Non-communicable diseases” in the last sentence of the background part of the abstract.

Response: thank you, for your important comment. Based on your recommendations, we have made revision of the manuscript accordingly

3. “Noncommunicable Diseases” is correct for keywords according to the MeSH database of PubMed. 

Response: We would like to say thank you once again for your fruitful comments. This has been addressed as your recommendation.

4. The first two sentences of the introduction, exploring the statistics, need to be explained which year the belong to, provided in the reference. 

Response: Based on your recommendations, we have addressed the issue.

In line 44-48.

5. A grammatical and linguistic edit is essential for this manuscript, as the numerous issues are apparent in it and make understanding the provided draft difficult.

Response: thank you very much for your important comment. We acknowledge that English is not our first language and we have edited the manuscript by an English language instructor. Based on your recommendations, we have made revision of the manuscript accordingly. 

6. 2nd paragraph of introduction: are NCDs comorbidities of sexual dysfunction or sexual dysfunction is a comorbidity of NCDs?

Response: Based on your recommendations, we have addressed the issue.

In line 55-56.

7. 4th paragraph of introduction: references for the first sentence should be merged in a citation manager.

Response: Based on your recommendations, we have addressed the issue.

In line 67-69.

8. The knowledge gap is not well addressed in the introduction of the manuscript and needs to be explained more.

Response: Based on your recommendations, we have addressed the issue.

In line 49-54.

9. The “Objective” and “Research question” after the introduction, could be merged into the introduction part and is a repeat of the mentioned notions in the introduction.

Response: Based on your recommendations, we have addressed the issue.

In line 73-77.

10. In methods and materials: in inclusion criteria, types of included NCDs should be provided specifically.

Response: Based on your recommendations, we have addressed the issue.

11. In methods and materials: in inclusion criteria, what is the reference for the standardized tool for sexual dysfunction measurement? Is it validated before for the Ethiopian population?

Response: Based on your recommendations, we have addressed the issue.

In line 95-102.

12. In methods and materials: in exclusion criteria, are there any criteria of exclusion for any of NCDs, participants, or sexual dysfunction disorders? The exclusion criteria could be more detailed based on the aims of this study.

Response: Based on your recommendations, we have addressed the issue.

13. In methods and materials: why didn’t authors search the Scopus database?

Response: We thank you very much for this important recommendation and comment. It was writing problems, so it is corrected accordingly

14. In methods and materials: the search period is stated to be from inception to 2020 and also a period of 2000-2020? Please clarify which one is the searched period exactly.

Response: We thank you very much. Included study published from 2000-2020

Searching time was from May 1-31/ 2020 and updated June 5/2020

15. In methods and materials: there is not so much information about the screening process of the articles and about how did the screening was done by one or two authors.

Response: Based on your recommendations, we have addressed the issue. 

In line 29-32.

16. The terms “gray literature” and “grey literature” are used both. Please, unify this term through the manuscript.

Response: Based on your recommendations, we have addressed the issue.

Gray literature is information produced by government agencies, academic institutions, and also the for-profit sector that is not typically made available by commercial publishers. Examples of gray literature include: Reports. Proceedings. Dissertations and theses.The term grey literature refers to research that is either unpublished or has been published in non-commercial form. Examples of grey literature include: government reports. policy statements and issues papers.

17. Results, Table 1, “type of measurement” column measures are not discussed in the Methods section and should be elaborated in detail and the issue of difference of measures among different studies should be addressed and discussed. How did the authors compare studies while the measure differed?

Response: Based on your recommendations, we have addressed the issue.

18. Subgroup analysis and sensitivity analysis explanations should be provided in the methods section instead of the results section.

 Response: Based on your recommendations, we have addressed the issue.

19. “Fig” and “Figure” terms should be unified as “Figure” in the manuscript.

Response: we did based on the comments 

20. Why there is no Figure 4 in or at the end of the manuscript?

 Response: we did based on the comments 

21. The terms “non-communicable patients” and “non-communicable clients” are not proper provided in many parts of the manuscript.

Response: Based on your recommendations, we have addressed the issue.

22. In discussion: did authors find any explanation for the higher prevalence of sexual dysfunction in patients with mental disorders? Please include some information about this important finding in the discussion section.

Response: Based on your recommendations, we have addressed the issue.

In line 219-223

23. In discussion: findings are not discussed well enough, and it seems authors could use literature to highlight the importance of the investigated notion and compare results with other similar studies. 

Response: Based on your recommendations, we have addressed the issue.

In line 219-223

24. Also, authors could replace the paragraph “Implications of the study findings” into the discussion as a part for policymakers and public health authorities, besides the provided implications for the clinicians.

Response: Based on your recommendations, we have addressed the issue.

25. Figure 5 legend, “log” should be corrected in the legend and “LogitSD” on the figure. 

 Response: Based on your recommendations, we have addressed the issue.

26. The six articles included in the final analysis and report is inconsistent in Table 1 and Supplementary Table 1 in comparison to Figures 2 and 3. “Gerensea H. et al. (2018)” and “Zewlde KH. et.al. (2017)” are in tables, but “Walle et al. (2018)” and “Seid et al. (2017)” in figures. 

Response: we did based on the comments

27. Included paper in analyses with reference numbers 15 “tesfaye t, bayisa m, mesfin N: Prevalence and associated factors of erectile dysfunction among men DM patients in Gondar university hospital, Gondar Ethiopia. In.: Research Square; 2020” and reference number 18 “Gebrezgabhier G, Desta H, Berhe T, Hailu E, Gebrehiwot F, Kifle Y: Prevalence and associated factors of Female Sexual Dysfunction among Female Population in Aksum Town, Tigray Region, Ethiopia, 2019. A Community Based Cross Sectional Study. In.: Research Square; 2019” included in reports are preprints in the Research Square database and not peer-reviewed articles. And this in contrast with your methods!

 Response: We thank you very much for this important recommendation. We consider both published and unpublise research work. We modify the method parts 

28. Included paper in analyses with reference numbers 16 “Gerensea H, Tarko S, Zenebe Y, Mezemir R, Walle B, Lebeta KR, Fita YD, Abdissa HG: Prevalence of erectile dysfunction and associated factors among diabetic men attending the diabetic clinic at Felege Hiwot Referral Hospital, Bahir Dar, North West Ethiopia, 2016. BMC endocrine disorders 2018, 11(1):130” is not traceable in online databases! 

Response: We thank you very much for this important recommendation. We correct like Bizuayehu Walle, Kidist Reba Lebet , Yamrot Debela Fitaand Hordofa Gutema Abdissa. Prevalence of erectile dysfunction and associated factors among diabetic men attending the diabetic clinic at Felege Hiwot Referral Hospital, Bahir Dar, North West Ethiopia, 2016. BMC endocrine disorders 2018, 11(1):130”

29. Included paper in analyses with reference numbers 17 “Zewlde KH, Muluneh NY, Seraj ZR, GebreLibanos MW, Bezabih YH, Seid A: Prevalence and determinants of erectile dysfunction among diabetic patients attending in hospitals of central and northwestern zone of Tigray, northern Ethiopia: a cross-sectional study. BMC neurology 2017, 17(1):16” is not traceable in online databases!

Response: We thank you very much for this important recommendation. We correct like Awole Seid , Hadgu Gerensea, Shambel Tarko, Yosef Zenebe and Rahel Mezemir. Prevalence and determinants of erectile dysfunction among diabetic patients attending in hospitals of central and northwestern zone of Tigray, northern Ethiopia: a cross-sectional study. BMC neurology 2017. 

30. Included paper in analyses with reference numbers 20 “Ejigu AK: Sexual dysfunction and associated factors among patients with epilepsy at Amanuel Mental Specialty Hospital, Addis Ababa - Ethiopia. Turkish journal of obstetrics and gynecology 2019, 19(1):255” is not published in the mentioned journal provided in the citation, instead of in the BMC Neurology journal!

 Response: We thank you very much, we checked and published at BMC Neurology journal

---

## [Decision Letter · Decision Letter 1]

12 Mar 2021

PONE-D-20-18974R1

Estimate the burden of sexual dysfunction due to non-communicable diseases in Ethiopia : systematic review and meta-analysis

PLOS ONE

Dear Dr. Eshete,

Thank you for submitting your manuscript to PLOS ONE. After careful consideration, we feel that it has merit but does not fully meet PLOS ONE’s publication criteria as it currently stands. Therefore, we invite you to submit a revised version of the manuscript that addresses the points raised during the review process.

We look forward to receiving your revised manuscript.

Kind regards,

Claudia Marotta

Academic Editor

PLOS ONE

Additional Editor Comments (if provided):

dear authors follow Reviewers suggestions to improve your paper

Reviewers' comments:

Reviewer's Responses to Questions

**Comments to the Author**

1. If the authors have adequately addressed your comments raised in a previous round of review and you feel that this manuscript is now acceptable for publication, you may indicate that here to bypass the “Comments to the Author” section, enter your conflict of interest statement in the “Confidential to Editor” section, and submit your "Accept" recommendation.

Reviewer #1: All comments have been addressed

Reviewer #2: (No Response)

Reviewer #3: (No Response)

2. Is the manuscript technically sound, and do the data support the conclusions?

Reviewer #1: Yes

Reviewer #2: Partly

Reviewer #3: No

3. Has the statistical analysis been performed appropriately and rigorously? 

Reviewer #1: Yes

Reviewer #2: Yes

Reviewer #3: I Don't Know

4. Have the authors made all data underlying the findings in their manuscript fully available?

Reviewer #1: Yes

Reviewer #2: Yes

Reviewer #3: Yes

5. Is the manuscript presented in an intelligible fashion and written in standard English?

Reviewer #1: Yes

Reviewer #2: No

Reviewer #3: No

6. Review Comments to the Author

Reviewer #1: Question 1

My comments in the previous review have been adequately taken care of, the objective of the study now agrees with the findings

Reviewer #2: Authors of this study revised the manuscript based on the provided comments. The changes were noticeable. However, some major points still remain to be answered and considered in the manuscript. A response to the authors’ letter is provided below, by number of previous comments of mine:

1. Comment well addressed. Title is edited.

2. Comment not addressed.

3. Comment well addressed.

4. Comment well addressed.

5. Comment for language edit is addressed. However a final round of language edit is necessary prior to publication.

6. Comment well addressed.

7. Comment well addressed.

8. Comment not addressed. I meant the knowledge gap about the sexual dysfunction and NCDs in the literature. Authors added an irrelevant part about knowledge on diabetes to the introduction!

9. Comment well addressed.

10. Comment well addressed.

11. Comment partially addressed. Please add the full terms of IIEF and CSFQ. Also, please add whether the tool is standardized for the Ethiopian population or not.

12. Comment not addressed.

13. Comment well addressed.

14. Comment partially addressed. Please add the exact time period of search to the text.

15. Comment not addressed. The screening process is different from the data extraction. This comment is a major one and still needs to be answered.

16. Comment not addressed. Please provide reference for the difference you mean between gray and grey literature.

17. Comment partially addressed. Types of measurement are added to the methods. But still authors did not answer how they compare studies when the measurement tool is different. As the man concern, how did authors conduct the meta-analysis when the measurement tools were different among the included studies? It is recommended to do meta in each category separately.

18. Comment not addressed.

19. Comment well addressed.

20. Comment well addressed.

21. Comment well addressed.

22. Comment well addressed.

23. Comment well addressed.

24. Comment well addressed.

25. Comment not addressed.

26. Comment partially addressed. Reference to “Walle (2018)” is corrected. Reference to “Seid (2017)” is incorrect in the bibliography. Also, statistics of mild, moderate, and severe SD in Table 1 are not consistent with numbers in the original paper! Also, data for “Ejigu (2019)” has a problem as the total prevalence of SD was 63.9 and is stated to be 63.3.

27. Comment well addressed.

28. Comment well addressed.

29. Comment partially addressed. Citation number 34 is still incorrect in the bibliography of manuscript. Please correct it as you answered the comment.

30. Comment not addressed. Please correct the citation in the bibliography f manuscript.

Reviewer #3: The language used throughout the manuscript is sub-standard and needs significant improvement. Several of the paragraphs are constructed poorly, words are used incorrectly, poor grammar is used, some paragraphs include very long incomprehensible sentences. Acronyms are used in the manuscript sometimes without explanation, and at other times after an acronym is used, the full term is reused in the manuscript., instead of the acronym. The manuscript contains repetition.

The Discussion is not cohesive or logical. Many loose statements are made with no references. The Conclusion of the manuscript’s includes a repetition of the study results.

7. PLOS authors have the option to publish the peer review history of their article (what does this mean?). If published, this will include your full peer review and any attached files.

Reviewer #1: **Yes: **Haruna Ismaila Adamu, MBBS; MPH; PhD

Reviewer #2: **Yes: **Sina Azadnajafabad MD MPH

Reviewer #3: No

---

## [Author Response · Author response to Decision Letter 1]

16 Apr 2021

Dear Academic Editor!

PLOS ONE 

Response to Reviewers

I am happy to resubmit for publication version of “Estimate the burden of sexual dysfunction due to non-communicable disease in Ethiopia: systematic review and meta-analysis” for a review as original research in PLOS ONE.

The comments of the editor and the reviewers were highly insightful and enabled us to greatly improve the quality of our manuscript. Therefore, based on the editor’s and the reviewers’ concerns we have made extensive edition in our manuscript. Particularly, we have extensively edited the manuscript by a professional language editor (English-language instructors of Debre Berhan University thoroughly edited the manuscript for language usage, spelling, and grammar) before submitting the revised version. The formatting of the text and document (text sizes and grammatical errors) was also edited. We have addressed yours’ concerns in a point by point format. 

 We look forward to hearing from you at your earliest convenience. 

Thank you for your consideration of this manuscript! 

Kind regards,

Akine Eshete

On behalf of authors

The authors of this study made a precious effort to investigate the burden of non-communicable diseases (NCDs) on sexual dysfunction in Ethiopia, especially focusing on the quality of life of patients with NCDs. This systematic review and meta-analysis tried to investigate the mentioned notion through a comprehensive review and statistical analyses. However, some comments are necessary to be addressed and I provide them below:

Reviewer #1: Question 1

My comments in the previous review have been adequately taken care of, the objective of the study now agrees with the findings

Reviewer #2: Authors of this study revised the manuscript based on the provided comments. The changes were noticeable. However, some major points still remain to be answered and considered in the manuscript. A response to the authors’ letter is provided below, by number of previous comments of mine:

1. Comment well addressed. Title is edited.

2. Comment not addressed (“Non-communicable disease” is better to be replaced by “Non-communicable diseases” in the last sentence of the background part of the abstract.)

Response: We thank you very much for this important recommendation. We edited it accordingly.

3. Comment well addressed.

4. Comment well addressed.

5. Comment for language edit is addressed. However a final round of language edit is necessary prior to publication.

Response: thank you very much for your important comment. We acknowledge that English is not our first language and we have edited the manuscript by an English language instructor. Based on your recommendations, we have made revision of the manuscript accordingly

6. Comment well addressed.

7. Comment well addressed.

8. Comment not addressed. I meant the knowledge gap about the sexual dysfunction and NCDs in the literature. Authors added an irrelevant part about knowledge on diabetes to the introduction!

Response: thank you very much for your important comment, we deleted

9. Comment well addressed.

10. Comment well addressed.

11. Comment partially addressed. Please add the full terms of IIEF and CSFQ. Also, please add whether the tool is standardized for the Ethiopian population or not.

Response: thank you very much for your important comment; we adopted already standardized tools based on the context of Ethiopia 

12. Comment well addressed.

13. Comment partially addressed. Please add the exact time period of search to the text (In methods and materials: the search period is stated to be from inception to 2020 and also a period of 2000-2020? Please clarify which one is the searched period exactly.)

Response: We thank you very much for this important recommendation. We edited on line 116 and 117

14. Comment not addressed. The screening process is different from the data extraction. This comment is a major one and still needs to be answered (In methods and materials: there is not so much information about the screening process of the articles and about how did the screening was done by one or two authors.) 

Response: We thank you very much for this important recommendation. We edited on line 129 and 136

15. Comment not addressed. Please provide reference for the difference you mean between gray and grey literature. 

Response: We thank you very much for this important recommendation. We edited on line 116

16. Comment partially addressed. Types of measurement are added to the methods. But still authors did not answer how they compare studies when the measurement tool is different. As the man concern, how did authors conduct the meta-analysis when the measurement tools were different among the included studies? It is recommended to do meta in each category separately.

Response: We thank you very much for this important recommendation; we understood that our outcome interest is sexual dysfunction. We summarized the prevalence of sexual dysfunction, which was assessed the same tool. 

17. Comment not addressed (Subgroup analysis and sensitivity analysis explanations should be provided in the methods section instead of the results section). 

Response: We thank you very much for this important recommendation. We edited on line from 207 to 209

18. Comment well addressed

19. Comment well addressed.

20. Comment well addressed.

21. Comment well addressed.

22. Comment well addressed

23. Comment well addressed

24. Comment not addressed (Figure 5 legend, “log” should be corrected in the legend and “LogitSD” on the figure.) 

Response: We thank you very much for this important recommendation. We edited it accordingly.

25. Comment partially addressed. Reference to “Walle (2018)” is corrected. Reference to “Seid (2017)” is incorrect in the bibliography. Also, statistics of mild, moderate, and severe SD in Table 1 are not consistent with numbers in the original paper! Also, data for “Ejigu (2019)” has a problem as the total prevalence of SD was 63.9 and is stated to be 63.3. 

Response: We thank you very much for this important recommendation. We edited it accordingly.

26. Comment well addressed.

27. Comment well addressed.

28. Comment partially addressed. Citation number 34 is still incorrect in the bibliography of manuscript. Please correct it as you answered the comment. It is corrected accordingly 

Response: We thank you very much for this important recommendation. We edited it accordingly.

29. Comment not addressed. Please correct the citation in the bibliography f manuscript. . It is corrected accordingly

Response: We thank you very much for this important recommendation. We edited it accordingly.

Reviewer #3: The language used throughout the manuscript is sub-standard and needs significant improvement. Several of the paragraphs are constructed poorly, words are used incorrectly, poor grammar is used, some paragraphs include very long incomprehensible sentences. Acronyms are used in the manuscript sometimes without explanation, and at other times after an acronym is used, the full term is reused in the manuscript., instead of the acronym. The manuscript contains repetition. The Discussion is not cohesive or logical. Many loose statements are made with no references. The Conclusion of the manuscripts includes a repetition of the study results.

Response: We thank you very much for these important comments. We corrected it accordingly the comments

---

## [Decision Letter · Decision Letter 2]

14 May 2021

PONE-D-20-18974R2

Estimate the burden of sexual dysfunction due to non-communicable diseases in Ethiopia : systematic review and meta-analysis

PLOS ONE

Dear Dr. Eshete,

Thank you for submitting your manuscript to PLOS ONE. After careful consideration, we feel that it has merit but does not fully meet PLOS ONE’s publication criteria as it currently stands. Therefore, we invite you to submit a revised version of the manuscript that addresses the points raised during the review process.

We look forward to receiving your revised manuscript.

Kind regards,

Vanessa Carels

Staff Editor

PLOS ONE

Journal Requirements:

Additional Editor Comments (if provided):

Reviewer 3 has noted ongoing concerns with the language quality throughout the submission. PLOS ONE requires that the language in submitted articles must be clear, correct, and unambiguous in order to meet our fifth criterion for publication (http://journals.plos.org/plosone/s/criteria-for-publication#loc-5), and PLOS ONE does not copy edit accepted manuscripts. Please revise the manuscript for English grammar and usage as well as for scientific readability. Please note that further consideration of this work requires that this criterion is met, please ensure your revision is thorough.  

Reviewers' comments:

Reviewer's Responses to Questions

**Comments to the Author**

1. If the authors have adequately addressed your comments raised in a previous round of review and you feel that this manuscript is now acceptable for publication, you may indicate that here to bypass the “Comments to the Author” section, enter your conflict of interest statement in the “Confidential to Editor” section, and submit your "Accept" recommendation.

Reviewer #1: All comments have been addressed

Reviewer #2: All comments have been addressed

Reviewer #3: (No Response)

2. Is the manuscript technically sound, and do the data support the conclusions?

Reviewer #1: Yes

Reviewer #2: Yes

Reviewer #3: Yes

3. Has the statistical analysis been performed appropriately and rigorously? 

Reviewer #1: Yes

Reviewer #2: Yes

Reviewer #3: Yes

4. Have the authors made all data underlying the findings in their manuscript fully available?

Reviewer #1: Yes

Reviewer #2: Yes

Reviewer #3: Yes

5. Is the manuscript presented in an intelligible fashion and written in standard English?

Reviewer #1: Yes

Reviewer #2: Yes

Reviewer #3: No

6. Review Comments to the Author

Reviewer #1: My observations in the previous reviews have been adequately addressed. I have no additional comments to make at this time.

Reviewer #2: Authors of this study revised the manuscript based on the provided comments. The changes were noticeable. The addressed comments and responses are acceptable. I hope a good production process for authors if other reviewers and dear editor approve the manuscript, too.

Reviewer #3: Review of PLOS Article Resubmission

The language used in the article is still sub-standard and needs to be improved. Some components of the manuscript are difficult to understand because of the poor sentence construction and grammar. Some sentences seem incomplete. An Acronym is introduced without explanation and then the full term is used in following text without use of the acronym. An Acronym is introduced and never used again in the text. The manuscript also contains long sentences which lack clarity.

Specifics are provided below:

Line 46 - The (WHO 2018) funding showed that 41 million cases and 57 million deaths - - Sentence unclear may be incomplete?

Line 47-49 - Sentences unclear - poor sentence construction and grammar

Lines 60-62 - long sentence and poor sentence construction

Lines 63-64 - incorrect grammar- “.....due to an “absent

The acronym SD is introduced in Line 63, but not used in the many other places following where sexual dysfunction is used (lines 68, 69, 70,73-74,78)

Lines 68-79 - unclear paragraph -

Lines 68-72 - is this about Ethiopia or worldwide (global)??

Lines 72-75 - incorrect grammar-...., this systematic review and meta-analysis were aimed to synthesize.....

Lines 76-79 - poor sentence construction and grammar

Lines 89-91 - incomplete sentence

Lines 94-100- Paragraph includes poor sentence construction and grammar

Lines 103-109 - unclear...Is that included as part of the Types of outcome measures?? Unclear what screening tool you are talking about..

Lines 118-120 - sentence incomplete

Line 124 - Acronym JBI used without explanation

Lines 124-127- Poor sentence construction

Line 129-135 - Poor grammar

Line 136- Unclear, poor sentence construction

Lines 141-150 - Poor paragraph/sentence construction, incomplete sentences, poor grammar

Lines 153-154 - incomplete sentence, incomprehensible

Lines 159 - 161 - poor sentence construction/grammar

Lines 171-172 - About 198 articles were excluded........ - Could the exact number of articles excluded be provided?..

Lines 175-176 - Poor grammar

Lines 195-197 - unclear paragraph

Lines 206-208 - To discover the potential reasons for the heterogeneity by subgroup meta-analyses were analyzed...awkward sentence/paragraph.

DISCUSSION

While improvements have been made to the Discussion by including references, the grammar and sentence construction still needs to be improved.

Lines 235-236 - poor grammar

Lines 239 - “sexual dysfunctions” - poor grammar

Line 240 - “complications was critical”- wrong tense

Lines 246-248 - poor sentence construction

Lines 251-252 - poor sentence construction

Lines 252-253 - poor sentence construction

Lines 257-259 - unclear sentence

Lines 262-266 - Very long unclear sentence

Lines 266-267 - unclear sentence

Lines 269 - 271 - poor sentence construction

Lines 274-279 - Poor construction of sentences and paragraph

Lines 280-281 - unclear sentence

Lines 285 -286 - It is “mandatory” to include screening and management ..... - Mandatory is a very strong word. - Please include reference.

Lines 294-302 - Poor grammar, very long sentence and tense confusion in the paragraph

Conclusion

The conclusion remains weak with results repeated. Requires revision.

7. PLOS authors have the option to publish the peer review history of their article (what does this mean?). If published, this will include your full peer review and any attached files.

Reviewer #1: **Yes: **Haruna Ismaila Adamu, MBBS; MPH; PhD

Reviewer #2: **Yes: **Sina Azadnajafabad, MD, MPH.

Reviewer #3: **Yes: **Glennis Andall-Brereton

---

## [Author Response · Author response to Decision Letter 2]

20 May 2021

Dear Academic Editor!

PLOS ONE 

Response to Reviewers

I am happy to resubmit for publication version of “Estimate the burden of sexual dysfunction due to non-communicable disease in Ethiopia: systematic review and meta-analysis” for a review as original research in PLOS ONE.

The comments of the editor and the reviewers were highly insightful and enabled us to greatly improve the quality of our manuscript. We have addressed yours’ concerns in a point by point format. 

 Thank you for your consideration of this manuscript! 

Kind regards,

Akine Eshet

Comments 

Journal Requirements:

Response: thank you, for your important comment. We check and all references are complete and correct in line with the journal requirement. 

Comments of reviewer 

1. Line 46 - The (WHO 2018) funding showed that 41 million cases and 57 million deaths - - Sentence unclear may be incomplete?

2. Line 47-49 - Sentences unclear - poor sentence construction and grammar

3. Lines 60-62 - long sentence and poor sentence construction

4. Lines 63-64 - incorrect grammar- “.....due to an “absent 

5. The acronym SD is introduced in Line 63, but not used in the many other places following where sexual dysfunction is used (lines 68, 69, 70,73-74,78)

6. Lines 68-79 - unclear paragraph –

Response: thank you, for your important comment. Based on your recommendations, we have made revision accordingly (Line 46-79)

7. Lines 68-72 - is this about Ethiopia or worldwide (global)??

Response: thank you, for your important comment. In the paragraph we mean that patients with non-communicable diseases in Ethiopia, we corrected 

8. Lines 72-75 - incorrect grammar-...., this systematic review and meta-analysis were aimed to synthesize.....

9. Lines 76-79 - poor sentence construction and grammar

10. Lines 89-91 - incomplete sentence

11. Lines 94-100- Paragraph includes poor sentence construction and grammar

Response: thank you, for your important comment. Based on your recommendations, we have made revision accordingly (line 72-100)

12. Lines 103-109 - unclear...Is that included as part of the Types of outcome measures?? Unclear what screening tool you are talking about.

Response: thank you, for your important comment. Types of outcome measures/ domain of sexual dysfunction/, finally after computing these parameterise, researchers categorize having sexual dysfunction and not. 

13. Lines 118-120 - sentence incomplete

14. Line 124 - Acronym JBI used without explanation

15. Lines 124-127- Poor sentence construction

16. Line 129-135 - Poor grammar

17. Line 136- Unclear, poor sentence construction

18. Lines 141-150 - Poor paragraph/sentence construction, incomplete sentences, poor grammar

19. Lines 153-154 - incomplete sentence, incomprehensible

20. Lines 159 - 161 - poor sentence construction/grammar

Response: thank you, for your important comment. Based on your recommendations, we have made revision accordingly (line 118-161)

21. Lines 171-172 - About 198 articles were excluded, Could the exact number of articles excluded be provided

Response: thank you, for your important comment. Using the developed Boolean operators any one checks and confirms the available studies, we excluded by considering the title and reading of abstracts 

22. Lines 175-176 - Poor grammar

23. Lines 195-197 - unclear paragraph

24. Lines 206-208 - To discover the potential reasons for the heterogeneity by subgroup meta-analyses were analyzed...awkward sentence/paragraph.

Response: thank you, for your important comment. Based on your recommendations, we have made revision accordingly (line 118-161)

DISCUSSION

While improvements have been made to the Discussion by including references, the grammar and sentence construction still needs to be improved.

25. Lines 235-236 - poor grammar

26. Lines 239 - “sexual dysfunctions” - poor grammar

27. Line 240 - “complications was critical”- wrong tense

28. Lines 246-248 - poor sentence construction

29. Lines 251-252 - poor sentence construction

30. Lines 252-253 - poor sentence construction

31. Lines 257-259 - unclear sentence

32. Lines 262-266 - Very long unclear sentence

33. Lines 266-267 - unclear sentence

34. Lines 269 - 271 - poor sentence construction

35. Lines 274-279 - Poor construction of sentences and paragraph

36. Lines 280-281 - unclear sentence

Response; Thanks this important comments, we did accordingly (line 235-281)

37. Lines 285 -286 - It is “mandatory” to include screening and management ..... - Mandatory is a very strong word. - Please include reference.

Response; Thanks this important comments, we cited 

38. Lines 294-302 - Poor grammar, very long sentence and tense confusion in the paragraph

Conclusion

39. The conclusion remains weak with results repeated. Requires revision.

Response; Thanks this important comments, we did accordingly

---

## [Decision Letter · Decision Letter 3]

30 Jun 2021

PONE-D-20-18974R3

Estimate the burden of sexual dysfunction due to non-communicable diseases in Ethiopia : systematic review and meta-analysis

PLOS ONE

Dear Akine Abosetugn Eshete,

I am writing to notify you regarding the submitted manuscript "Estimate the burden of sexual dysfunction due to non-communicable diseases in Ethiopia : systematic review and meta-analysis". As an act of transparency, I am invited to handle your paper as a guest academic editor, after multiple rounds of commenting and reviewing your paper in the previous stages. After careful consideration and inspection based on the rounds of revision and the current status of your manuscript, I decided to invite one of the previous reviewers and two new expert reviewers to comment on your manuscript, as it still lacks some details and does not fully meet PLOS ONE’s publication criteria as it currently stands. However, the comments lead to a minor revision decision. Therefore, I invite you to submit a revised version of the manuscript that addresses the points raised during the recent review process.

We look forward to receiving your revised manuscript.

Kind regards,

Sina Azadnajafabad

Academic Editor

PLOS ONE

Journal Requirements:

Additional Editor Comments (if provided):

Dear authors,

Please follow Reviewers suggestions to revise your paper.

Reviewers' comments:

Reviewer's Responses to Questions

**Comments to the Author**

1. If the authors have adequately addressed your comments raised in a previous round of review and you feel that this manuscript is now acceptable for publication, you may indicate that here to bypass the “Comments to the Author” section, enter your conflict of interest statement in the “Confidential to Editor” section, and submit your "Accept" recommendation.

Reviewer #3: All comments have been addressed

Reviewer #5: (No Response)

2. Is the manuscript technically sound, and do the data support the conclusions?

Reviewer #3: Yes

Reviewer #4: Yes

Reviewer #5: Yes

3. Has the statistical analysis been performed appropriately and rigorously? 

Reviewer #3: Yes

Reviewer #4: Yes

Reviewer #5: Yes

4. Have the authors made all data underlying the findings in their manuscript fully available?

Reviewer #3: Yes

Reviewer #4: Yes

Reviewer #5: Yes

5. Is the manuscript presented in an intelligible fashion and written in standard English?

Reviewer #3: Yes

Reviewer #4: Yes

Reviewer #5: Yes

7. PLOS authors have the option to publish the peer review history of their article (what does this mean?). If published, this will include your full peer review and any attached files.

Reviewer #3: **Yes: **Glennis Andall-Brereton

Reviewer #4: **Yes: **Esmaeil Mohammadi, MD MPH

Reviewer #5: **Yes: **Mohammad Keykhaei

6. Review Comments to the Author

Reviewer #4: I wanted to first thank the editorial board for the opportunity of reviewing this paper. It is a SRMA with title of ‘Estimate the burden of sexual dysfunction due to non-communicable diseases in Ethiopia: systematic review and meta-analysis'. It seems that the authors and prior reviewers have made a substantial effort to reach to this well-revised paper. The introduction has been designed accordingly to talk about the condition, gaps, and aims of project. Besides identification of abbreviations for their first appearance (e.g., IIEF-5), other parts of the methods was well organized based on the PICO. Based on the journal of preference, search string can be exported to the appendix. Table 1 needs abbreviations and explanations independent of main txt and all should be mentioned at footnote (e.g., SNNPR, ED...). I believe the first sentence of line 190 is not complete or it is just vague. Otherwise, the results sections are prepared well. The PRISMA chart is excrutiatingly messy and inconsistent. Please revise this part and unify numbers with the text. Authors have profoundly discussed their findings and have no comment on this part, though the conclusion can be further elaborate the need to understanding SD in NCD patients, so on so forth.

Reviewer #5: Dear Authors,

I read the manuscript entitled " Estimate the burden of sexual dysfunction due to non-communicable diseases in Ethiopia: systematic review and meta-analysis”. The paper is well written, and both the topic and data analysis are impressive, although attention to the following issues could improve the quality of the paper.

Minor comments

Introduction section:

1. Line 48, “of from” is not grammatically correct.

2. Line 49 “It” must be substituted with "NCDs".

3. Line 54, “Hypertension” should not be capitalized.

Methods:

1. The conducted search strategy is up to June 2020. This strategy should be updated.

2. In Figure 1, in the screening section, the authors have identified that 198 studies have been excluded. The exclusion might be due to some reasons including “review article”, “not relating to the subject” and etc. So, these reason with the numbers of articles in each exclusion section should be written in this fellow chart.

Discussion:

1. The authors should add additional paragraphs regarding the solutions for preventing or screening sexual dysfunction.

2. The authors have not discussed the reasons and potential solutions for sexual dysfunction in patients with diabetes. This part should be included.

---

## [Author Response · Author response to Decision Letter 3]

22 Sep 2021

Dear Academic Editor!

PLOS ONE 

Response to Reviewers

I am pleased to resubmit the revised version of "Estimate the burden of sexual dysfunction related to non-communicable disease in Ethiopia: systematic review and meta-analysis" to PLOS ONE for publication as original research.

The editor's and reviewers' comments were quite helpful, allowing us to significantly increase the quality of our article. As a result, we have made considerable revisions to our paper in accordance to the editor's and reviewers' comments. 

Before submitting the revised version, the manuscript extensively edited by a professional language editor (English-language instructors of Debre Berhan University thoroughly edited the manuscript for language usage, spelling, and grammar). The text and document's formatting (text sizes and grammatical errors) were also edited. Your concerns have been addressed in a point-by-point approach.

 We look forward to hearing from you at your earliest convenience. 

Thank you for your consideration of this manuscript! 

Kind regards,

Akine Eshete

On behalf of authors

The authors of this study made a precious effort to investigate the burden of non-communicable diseases (NCDs) on sexual dysfunction in Ethiopia, especially focusing on the quality of life of patients with NCDs. This systematic review and meta-analysis tried to investigate the mentioned notion through a comprehensive review and statistical analyses. However, some comments are necessary to be addressed and I provide them below:

Reviewer #4

Besides identification of abbreviations for their first appearance (e.g., IIEF-5)

Response: We thank you very much for this important recommendation. We edited it accordingly on line 91

Table 1 needs abbreviations and explanations independent of main txt and all should be mentioned at footnote (e.g., SNNPR, ED...). 

Response: We thank you very much for this important recommendation. We edited it accordingly on line 170

I believe the first sentence of line 190 is not complete or it is just vague.

Response: We thank you very much for this important recommendation. We edited it accordingly on line 182.

The PRISMA chart is excruciatingly messy and inconsistent. Please revise this part and unify numbers with the text.

Response: We thank you very much for this important recommendation. We tried to modify based on the comments. 

Reviewer #5

1. Line 48, “of from” is not grammatically correct.

Response: We thank you very much for this important recommendation. We edited it accordingly from line 47-48.

2. Line 49 “It” must be substituted with "NCDs".

Response: We thank you very much for this important recommendation. We edited it accordingly

3. Line 54, “Hypertension” should not be capitalized.

Response: We thank you very much for this important recommendation. We edited it accordingly on line 54 

Methods:

1. The conducted search strategy is up to June 2020. This strategy should be updated.

Response: We thank you very much for this important recommendation. We edited it accordingly

2. In Figure 1, in the screening section, the authors have identified that 198 studies have been excluded. The exclusion might be due to some reasons including “review article”, “not relating to the subject” and etc. So, these reason with the numbers of articles in each exclusion section should be written in this fellow chart.

Response: We thank you very much for this important recommendation. We edited it accordingly from line 157-158

Discussion:

1. The authors should add additional paragraphs regarding the solutions for preventing or screening sexual dysfunction.

Response: We thank you very much for this important recommendation. We edited it accordingly from line 285-310

2. The authors have not discussed the reasons and potential solutions for sexual dysfunction in patients with diabetes. This part should be included.

Response: We thank you very much for this important recommendation. We edited it accordingly from line 285-310

---

## [Editor Report · Decision Letter 4]

11 Oct 2021

Estimate the burden of sexual dysfunction due to non-communicable diseases in Ethiopia : systematic review and meta-analysis

PONE-D-20-18974R4

Dear Dr. Akine Abosetugn Eshete,

We’re pleased to inform you that your manuscript has been judged scientifically suitable for publication and will be formally accepted for publication once it meets all outstanding technical requirements.

Kind regards,

Sina Azadnajafabad

Guest Editor

PLOS ONE

Additional Editor Comments (optional):

Dear Dr. Akine Abosetugn Eshete,

I am writing to you regarding the manuscript "Estimate the burden of sexual dysfunction due to non-communicable diseases in Ethiopia : systematic review and meta-analysis". As an act of transparency, I am invited to handle your paper as a guest academic editor for the second time, after multiple rounds of commenting and reviewing of your paper in the previous stages.

After careful consideration and inspection based on the recent round of review by two additional reviewers, and the following revision and resubmission by authors, I reached the decision of accept for the manuscript in the current format of it. The authors made the necessary changes to the manuscript and it is suitable for publication in PLOS One now. I do appreciate the effort of all editors, reviewers, and authors of this paper in the process of evaluation.

Sina Azadnajafabad, MD MPH

---

## [Editor Report · Acceptance letter]

19 Oct 2021

PONE-D-20-18974R4 

Estimate the burden of sexual dysfunction due to non-communicable diseases in Ethiopia: systematic review and meta-analysis 

Dear Dr. Eshete:

I'm pleased to inform you that your manuscript has been deemed suitable for publication in PLOS ONE. Congratulations! Your manuscript is now with our production department. 

Kind regards, 

on behalf of

Dr. Sina Azadnajafabad 

Guest Editor

PLOS ONE